# Effects of Biostimulants on the Eco-Physiological Traits and Fruit Quality of Black Chokeberry (*Aronia melanocarpa* L.)

**DOI:** 10.3390/plants13213014

**Published:** 2024-10-28

**Authors:** Anastasia Giannakoula, Georgia Ouzounidou, Stefanos Stefanou, George Daskas, Olga Dichala

**Affiliations:** 1Department of Agriculture, International Hellenic University, 57400 Sindos, Greece; stefst2@ihu.gr (S.S.); olga.dichala@gmail.com (O.D.); 2Institute of Technology of Agricultural Products, ELGO-DIMITRA, 14123 Lycovrissi, Greece; 3Geogreen Marathon, 235-237 Marathonos Street, 19007 Marathonas, Greece; geogreenmarathon@gmail.com

**Keywords:** Aronia, biostimulants, antioxidant capacity, nutritional status, ascorbic acid, photochemical efficiency

## Abstract

Biostimulants contribute to the physiological growth of plants by enhancing the quality characteristics of fruit without harming the environment. In addition, biostimulants applied to plants strengthen nutritional efficiency, abiotic stress tolerance, and fruit biochemical traits. We investigated the effectiveness of specific organic biostimulants. Five treatments were tested: (1) control (H_2_O, no biostimulants); (2) Magnablue + Keyplex 350 (Mgl + Kpl350); (3) Cropobiolife + Keyplex 120 (Cpl + Kpl120); (4) Keyplex 120 (Kpl120); and (5) Magnablue + Cropobiolife + Keyplex 120 (Mgl + Cpl + Kpl120) on the mineral uptake and physiology in black chokeberry (Aronia) plants, as well as the quality of their berries. The different treatments were applied to three-year-old chokeberry plants, and the experimental process in the field lasted from May to September 2022 until the harvest of ripe fruits. Inductively Coupled Plasma Optical Emission Spectroscopy (ICP-OES) revealed that the fifth treatment significantly increased concentrations of P, Ca, and K. Additionally, the fifth treatment enhanced photochemical efficiency (Fv/Fm), water-splitting efficiency (Fv/Fo) in PSII, and the performance index (PI) of both PSI and PSII in chokeberry leaves. Improvements in photosynthesis, such as CO_2_ assimilation (A), transpiration (E), and water-use efficiency (A/E), were also noted under biostimulant applications. Upon harvesting the ripe fruits, part of them was placed at room temperature at 25 °C, while the rest were stored at 4 °C, RH 90% for 7 days. The cultivation with biostimulants had beneficial effects on the maintenance of flesh consistency, ascorbic acid concentration, and weight of berries at 4 and 25 °C, especially in the 5th treatment. Moreover, the total antioxidant capacity, anthocyanin concentration, and total phenols of the berries were notably higher in the third and fifth treatments compared to the control. These data suggest that selecting appropriate biostimulants can enhance plant yield and fruit quality by potentially activating secondary metabolite pathways.

## 1. Introduction

Black chokeberry (*Aronia melanocarpa* L.), a shrub of the Rosaceae family known for its nutrient-rich berries, is particularly valued for its health benefits, earning it a place among the superfoods [1]. Black chokeberry is a fruit that accumulates an abundance of nutrients that are beneficial for human health. A comparative study using the oxygen radical absorbing capacity assay showed that acetone extracts from black chokeberries exhibited stronger antioxidant activity than those obtained from blueberries (*Vaccinium corymbosum* L.) (over five times), cranberries (*Vaccinium macrocarpon* L.) (over eight times), and lingonberries (*Vaccinium vitisidaea* L.) (over four times). Most literature data concerning the chemistry of *A. melanocarpa* L. refer to its berries being a rich source of pharmacologically relevant compounds [2,3]. Polyphenols, especially anthocyanins and procyanidins, make up the main group of biologically active constituents in black chokeberry fruits [4,5]. High anthocyanin contents in chokeberries led to intensive scientific research into the biological activities of Aronia extracts [6], as much attention has been recently given to the chemopreventive action of polyphenols, as well as to their role in alleviating the symptoms of many human diseases. In chokeberry fruits, anthocyanins are the second largest group of phenolic compounds, with a concentration range from 0.60% to 2.00% dry weight. Anthocyanins present in black chokeberry are a mixture of cyanidin glycosides [7].

Through its new European Green Deal, announced in 2021, the European Union (EU) wants to lead the world on climate action and promote sustainable agriculture in a low-carbon economy. As part of this plan and in a bid to reduce nutrient and soil fertility loss by at least 50%, the 27-country bloc will cut overall chemical pesticide use in half and decrease fertilizer use by a minimum of 20% by 2030 [8]. In this frame, the use of biostimulants in agriculture has garnered significant attention due to their potential to enhance plant growth, improve crop quality, and mitigate environmental stressors without harmful side effects [9]. Biostimulants are a diverse category of substances and microorganisms that, when applied to plants, seeds, or soil, enhance plant growth, development, and resilience. Unlike fertilizers, which provide nutrients, or pesticides, which protect against pests and diseases, biostimulants work by stimulating natural processes in plants, improving their overall health and productivity [10]. The most common types of biostimulants are humic substances, seaweed extracts, microbial inoculants [11], amino acids, vitamins, and antioxidants. Biostimulants operate through several mechanisms, such as enhancing nutrient uptake, stimulating hormonal activity, improving stress tolerance, and promoting soil health [12]. The benefits of biostimulants are, firstly, the improved yield and quality [13] and, secondly, the environmental sustainability as well as the enhanced resilience [14]. This study explores the impact of specific organic biostimulants on the growth, physiological characteristics, and fruit quality of *Aronia melanocarpa* L. Considering the components of four commercial biostimulants, we designed experiments aimed at evaluating the response of certain parameters such as chlorophyll fluorescence, photosynthetic process, antioxidant capacity and flesh texture of the fruits compared to control plants. In addition, by investigating various treatments, an effort has been made to elucidate how these biostimulants can facilitate nutrient acquisition by supporting metabolic processes and improving the overall performance and resilience of black chokeberry plants, offering valuable insights for sustainable agricultural practices.

## 2. Results

### 2.1. Chlorophyll Fluorescence

The maximum photochemical yield (Fv/Fm), the water splitting efficiency (Fv/Fo) of photosystem II (PSII), and the performance index (PI) of photosystem I (PSI) and PSII in Aronia leaves were increased in the fifth treatment (Mgl + Cpl + Kpl120) (by 2, 9, and 42% with respect to the first, respectively) (Figure 1). The lower values were recorded in the third (Cpl + Kpl120) and fourth (Kpl120) treatments. Additionally, PI was the highest for the fifth and the lowest for the fourth treatment.

### 2.2. Photosynthesis

CO_2_ assimilation (A), transpiration (E), and water use capacity (WUE = A/E) changed under the action of the biostimulants, with a maximum WUE increase of 70% in the second treatment compared to the first (Figure 2). The application of Magnablue containing copper nanoparticles enhanced the photosynthetic activity and more efficient water use by the plants.

### 2.3. Weight Loss

The weight loss rate is a crucial metric for evaluating fresh-cut fruit quality. After 7 days of fruit harvest and storage at 4 °C or 25 °C, significant fluctuations have been observed in the chokeberry biomass. Control fruits stored at 25 °C for 7 days revealed a significantly greater weight loss by about 68% on the first day of harvest. When exposed to the fifth treatment (Mgl + Cpl + Kpl120) and stored at 25 °C, the biomass loss was 60%. Notably, the storage at 4 °C and the cultivation with the fifth (Mgl + Cpl + Kpl120) treatment exhibited consistently lower weight loss rates (by about 10%) than the control and other biostimulant treatments (Figure 3). The use of the specific biostimulants, as well as the storage at 4 °C for 7 days, effectively delayed the weight loss of fresh-cut chokeberry fruits.

### 2.4. Texture

Exposure to different biostimulants contributes to maintaining the flesh texture of chokeberry fruits, which was similar to all treatments studied at day 0 (Figure 4). The storage at 4 °C and 25 °C for 7 days caused significant variation in fruit flesh resistance. The flesh resistance of the control fruits and fruits treated with the fourth treatment (Kpl120) were significantly reduced by 48% and 22% compared to day 0 after storage to 25 °C, respectively (Figure 4), since the higher force was needed to penetrate the fruit. In general, the use of biostimulants induced an increase in fruit integrity after 7 days of storage at 4 °C. The higher fruit flesh resistance has been shown on cultivation to the fifth treatment (Mgl + Cpl + Kpl120) by about 20% compared to the relevant value of day 0 (Figure 4). These results match those of the weight loss. It seems that the presence of certain biostimulants helps fruits maintain integrity and structure, while others lead to an acceleration of tissue aging, resulting in higher weight loss and firmness.

### 2.5. Ascorbic Acid

Fruits contain the highest amount of ascorbic acid (AA) when they are in their fresh activated state, while any processing handling tends to decrease the original content of this molecule. In our study, the highest amount of ascorbic acid was measured for all the treatments studied on day 0; more precisely, AA production increased along with the third (Cpl + Kpl120) and fifth (Mgl + Cpl + Kpl120) biostimulant applications by 10 and 7% compared to the untreated fruits, respectively (Figure 5). In fruits stored at 25 °C for 7 days, a significant loss of AA concentration was observed; under the fourth (Kpl120) treatment, a reduction of 53% on day 0 was measured. Storage at 4 °C caused maintenance of AA content, showing the smallest decrease after the fifth treatment of 10% of the day 0 value (Figure 5).

### 2.6. Anthocyanins, Total Phenolic Content, and Total Antioxidant C Apacity

Anthocyanin content measured at day 0 in chokeberries was significantly affected by the application of biostimulants compared to the untreated fruits (Figure 6). The maximum value was detected after the fifth treatment (Mgl + Cpl + Kpl120) (by 33% of the control) and the lowest one after the fourth treatment. A similar trend was observed for the total phenolic content (TPC) in fruits, which showed the greatest values under the fifth treatment (Mgl + Cpl + Kpl120), which was 38% higher than the control. Biostimulants markedly enhanced the antioxidant capacity of the chokeberry fruits. All DPPH (1,1-diphenyl-2-picrylhydrazyl) values were higher than those of the untreated plants, giving the greatest values in the fifth treatment (Mgl + Cpl + Kpl120), by 32% of the control (Figure 6).

### 2.7. Elemental Composition

Chokeberry plants grown under the four types of biostimulants generally showed an increase in nutrient concentrations compared to the untreated plants (Figure 7, Figure 8, Figure 9 and Figure 10). The phosphorus (P) content in both leaves and berries shows significant variation across treatments. The control group shows the lowest P content in both leaves and berries. More precisely, a significant increase in P accumulation in leaves and fruits was observed in the second (Mgl + Kpl350), third (Cpl + Kpl120), and fifth (Mgl + Cpl + Kpl120) treatments, which were about three- and two-fold increases, respectively (Figure 7). Application of the fourth treatment caused increases to a lesser extent in P concentration in plant tissues.

A trend of increasing potassium (K) content in both leaves and fruits was observed under the biostimulants. Cultivation under the second (Mgl + Kpl350) and fifth (Mgl + Cpl + Kpl120) treatments induces the major augmentation three-fold (40%) and four-fold (80%) for leaves and fruits, respectively (Figure 8). Potassium levels in leaves were notably higher than in berries across all treatments, with the control group having the lowest K concentrations. The fifth (Mgl + Cpl + Kpl120) treatment significantly increased potassium content in both leaves and berries, with the leaves showing a particularly dramatic increase.

Similarly, calcium (Ca) concentration significantly increased under the biostimulant cultivation, showing an over-doubling trend in the fifth treatment (Mgl + Cpl + Kpl120) compared to the untreated plants (Figure 9). It can be noticed that the Ca content of fruits was higher in all the treatments than that of leaves.

Magnesium levels showed less variability across treatments compared to other nutrients, with leaves generally having slightly higher Mg concentrations than berries. The fifth (Mgl + Cpl + Kpl120) treatment resulted in the highest Mg content in both tissues, reinforcing the positive effect of combined biostimulants (Figure 10).

## 3. Discussion

EU regulation distinguishes between fertilizers and biostimulants, specifying how biostimulant activity manifests itself beyond the simple supply of nutrients to the plant. They are products based on natural extracts or beneficial microorganisms that improve crop response. They help the plant resist stresses such as high temperatures, drought, and soil salinity, which can support growth and productivity even in sub-optimal conditions. According to EBIC (European Biostimulants Industry Council) data, biostimulants can improve the absorption and use of nutrients by up to 25% and the intrinsic characteristics of the product (size, color, ripeness, impact resistance) by up to 15%. In addition, the European Green Deal presents a unique opportunity for biostimulant manufacturers to help the agricultural industry reduce synthetic chemical and fertilizer use as it works to feed the world [15].

In the present study, we looked into the potency of particular organic biostimulants in order to improve the physiological growth of plants and fruit quality attributes compared to untreated plants. The mixed biostimulants treatment enhanced the maximum efficiency of PSII photochemistry (Fv/Fm) of the chokeberry leaves as well as their functionality of the oxygen-evolving complex (OEC, Fv/Fo) [16], suggesting that this combination of biostimulants significantly improves the health and efficiency of the photosynthetic apparatus. Previously, studies also discussed how biostimulants influenced chlorophyll fluorescence and photosynthesis through enhanced nutrient availability [17,18]. The exposure to the Magnablue biostimulant had a dominant effect on CO_2_ assimilation and water use efficiency, showing higher values under the second and fifth treatments. The nano-copper, which is the main component of this biostimulant, strengthens plant photosynthesis and leads to advanced agricultural productivity. Similar results to our findings were reported by many earlier authors who observed the positive effect of other biostimulants on photosynthetic parameters [19,20]. In addition, the Performance Index (PI), which describes the overall expression of the plant’s internal strength in dealing with environmental conditions, shows significant enhancement under the mixture of the three biostimulants. Based on our results, the three functional steps of photosynthetic activities by the Reaction Center PSII complex such as light energy absorption, excitation energy absorption, and the conversion of absorbed energy to electron transport in PSII, can be improved by using specific doses and biostimulants.

A positive impact of the cold storage and application of biostimulant was observed on fruit quality. Indeed, fruits stored at 4 °C and under the mixture of the biostimulants revealed higher flesh resistance and lower weight loss. In addition, the firmness of the berries decreased across all treatments after 7 days at 25 °C, which is expected due to higher metabolic rates at elevated temperatures. Changes take place in the structure of the polysaccharides of the cell walls (pectins, hemicelluloses, celluloses), influenced by several preharvest treatments that contribute to maintaining fruit firmness for several weeks after harvest. Similar experiments with salicylic acid support the idea that biostimulants, particularly those that enhance antioxidant capacity, contribute to maintaining firmness during storage [21]. A decrease in the strength of the fruit’s skin during storage at high temperatures and ripening can be associated with the metabolic pathways that are responsible for textural changes in fruits, which are believed to involve loss in turgor pressure, degradation, and other physiological changes in the composition of membranes, degradation of starch, and modifications in the cell wall structure and dynamics [22,23,24].

In our study, the observation of effective preservation of berry firmness, possibly due to improved cell wall stability and delayed ripening processes under the combined treatment of biostimulants (Mgl + Cpl + Kpl120), may be due mainly to the presence of CropBioLife, which, based on activated flavonoids, contributed to the maintenance of the structure of the cell wall of the flesh of the fruits, the reduction in their weight loss, as well as the increase in the vitamin C of the fruits.

Moreover, calcium, in particular, is well-known for its role in maintaining cell wall integrity and firmness, as demonstrated by previous studies [25]. Previous findings have shown that biostimulants, particularly those that enhance antioxidant capacity, contributed to maintaining firmness during storage [26,27].

Regarding the total antioxidant capacity, concentration of anthocyanins, and total phenols of the fruits, the presence of Keyplex plays a decisive role. This commercial product containing specific defense proteins, ketone acids, and humic and fulvic acids mainly enhanced the Acquired Intersystemic Resistance, so plants revealed a significant increase mainly under the third and fifth handlings compared to the control. A lot of studies have also discussed how biostimulants and plant growth regulators could enhance secondary metabolite production, similar to the increase in antioxidants (anthocyanins, TPC, and DPPH) [28,29]. The stimulation of germination, seedlings, and plant growth, as well as crop productivity in response to biostimulant application, has been usually associated with the action of signaling bioactive molecules in the primary and secondary metabolisms [30,31].

Previous researchers [32] attributed the positive impact of the use of bioactive natural substances to enhance soil nutrient availability, plant nutrient uptake, and assimilation. Increasing nutrient use efficiency, in particular N and P, is fundamental for both economic and environmental reasons. Indeed, in our study, the elemental analysis showed that the fifth (Mgl + Cpl + Kpl120) treatment exhibited the highest phosphorus content, especially in the leaves, indicating that the combined biostimulants significantly enhanced phosphorus uptake. The trend suggests that the use of multiple biostimulants may promote a synergistic effect in phosphorus accumulation, particularly in leaves, with the berries also showing increased P content, though to a lesser extent. Furthermore, the fifth (Mgl + Cpl + Kpl120) treatment enhances potassium uptake and translocation within the plant. This result is important because potassium plays a crucial role in regulating stomatal function, water-use efficiency, and overall plant stress resistance, suggesting that this treatment could improve plant performance under stress conditions. Also, calcium levels were generally higher in the berries compared to leaves across most treatments, with the fifth treatment again leading to the highest calcium content in both tissues. This trend indicates that biostimulants, particularly in the combined treatment, enhanced calcium uptake, which was vital for maintaining cell wall structure and membrane stability. The higher Ca content in the berries may also improve their firmness and shelf life, contributing to better post-harvest quality [23,24]. The control treatment resulted in the lowest Ca levels, particularly in leaves, further highlighting the positive impact of the biostimulant treatments on nutrient accumulation.

Moreover, magnesium levels showed less variability across treatments compared to other nutrients, with leaves generally having slightly higher Mg concentrations than berries. The fifth treatment resulted in the highest Mg content in both tissues, reinforcing the positive effect of combined biostimulants. Magnesium is crucial for chlorophyll production and photosynthesis, so the increased Mg levels, particularly in leaves, suggest enhanced photosynthetic capacity in plants treated with biostimulants. The control treatment had the lowest Mg content, implying that the untreated plants may have experienced some level of deficiency that could negatively affect their overall growth and development. Significant increases in several fruit qualities, as well as mineral composition (P, K, Ca, Mg, Fe, Mn, and Zn), have been reported [30,33].

Our findings are in line with the results of [27,28], indicating improved nutrient availability and plant uptake efficiency, which can be attributed to the biostimulant treatments.

## 4. Materials and Methods

### 4.1. Experimental Design

This experiment was conducted at the farm of Alexander International Hellenic University (22°55′ E, 40°38′ N) during the period from March to September 2022.

Five different treatments were established with commercial biostimulants in three-year-old Aronia plants Control plants were irrigated with tap water; while biostimulants were diluted in tap water. The five treatments are depicted in Table 1.

Four replications were carried out by watering the roots of the plants every ten days, from May to September 2022, until the harvest of ripe fruits. Part of them was placed at room temperature at 25 °C, while the rest were placed in a preservation chamber at 4 °C, RH 90% for 7 days.

The concentrations of the Biostimulants used were selected from preliminary experiments, and some information about them is given below:

KeyPlex 350^®^ is a formulation of micronutrients most often found deficient in commercial crops and trees. It contains alpha-keto acids, which facilitate the utilization of micronutrients and increase resistance to environmental stress. Other micronutrients are Fe: 3.5%, Mn: 0.75%, Zn: 0.75%, B: 0.16%, Mg: 1.5%, and Mo: 0.003% *w*/*w* (https://www.keyplex.com/product/keyplex-350, accessed on 21 July 2017).

KeyPlex 120^®^ is the formulation of micronutrients (Fe: 0.2%, Mn: 0.11%, Zn: 0.10% *w*/*w*) availability most often found deficient in plants. It also contains humic acid, which may enhance soil micronutrient KP 120 is formulated for foliar application or fertigation to prevent and correct micronutrient deficiencies when used as directed (https://www.keyplex.com/product/keyplex-120, accessed on 26 October 2017).

Magnablue^®^ is a copper sulfate pentahydrate acid (Cu:5%, S: 4.27% *w*/*w*) product with systemic properties. It is a powerful systemic copper that can penetrate inside the leaves through stomata, thus activating a lot of enzyme procedures, enhancing photosynthesis, lignin production, and quicker absorption of nutrients, resulting finally in drastic empowerment of the plant’s defense mechanism (magnablue.gr, accessed on 20 December 2017).

CropBioLife’s power comes from its unique blend of activated flavonoids, while other dominant nutrients are N:1860 ppm, K:1370 ppm, S:482 ppm, and Na:720 ppm. The cycle of carbon removed from the air and deposited in the soil is the building block of regenerative agriculture. The flavonoids in CropBioLife^®^ work by improving the plant’s metabolism. This triggers an energy boost that enables efficiency in two very important plant functions. Photosynthesis and Root Exudation. The beginning of plant health starts with photosynthesis. By enabling efficient photosynthesis, we can observe a more significant carbon dioxide intake, promoting exponential energy production. The additional energy in the form of carbohydrates enables the plant to increase root exudation, which nurtures the microbial biology in the soil, enabling the plant to cultivate the microbes that provide the nutrients key to the plant’s survival (https://www.cropbiolife.com, accessed on 10 July 2019).

### 4.2. In Vivo Chlorophyll Fluorescence Measurements

In vivo chlorophyll fluorescence was measured on the upper leaf surface of the third from the top fully expanded leaf, using a Plant Analyser (PEA, Hansatech Ltd., King’s Lynn, Norfolk, UK) and parameters Fv/Fm, maximum efficiency of PSII photochemistry, Fv/Fo, the efficiency of the oxygen-evolving complex on the donor side of PSII, PI, and performance index were determined [34]. Measurements were conducted at 23 °C on intact leaves of four replicate plants from the five treatments.

### 4.3. Photosynthesis Measurements

Gas exchange was measured on the third from the top fully expanded leaf with an IR-GA Li-6400 portable photosynthesis meter (LiCor, Inc., Lincoln, NE, USA). Calculations of net photosynthetic rate (A), transpiration rate (E), water use efficiency (WUE = A/E), inter-cellular CO_2_ concentration (Ci), and stomatal conductance (gs) from gas exchange measurements were conducted [34]. Leaf temperature was about 25 °C; the relative humidity (RH) was 60–70%; CO_2_ concentration was 400 μL L^−1^, and light intensity was 1000 μmolm^−2^ s^−1^.

### 4.4. Determination of Weight Loss

The weight loss of the chokeberry fruits was measured after 7 days of harvest, during their storage at 4 °C and 25 °C. For the 4 °C storage, the weight was measured after temperature equilibrium at 25 °C. The weight loss was calculated through the difference in the initial weight and the weight at the sampling time of 10 fruits and expressed as a percentage of the initial weight of the fruits [35].

### 4.5. Texture Analysis

The flesh resistance of chokeberry fruits was measured using a Texture Analyzer (TA. HD plus-Stable Micro Systems Ltd., Surrey, UK) with a needle probe diameter of 2 mm [36]. The probe penetrated the flesh at 2 mm s^−1^ to a depth of 2.5 mm, measuring flesh resistance in Newtons (N). A total of nine samples per treatment were analyzed immediately after harvest and at 4 °C and 25 °C storage for 7 days after harvesting.

### 4.6. Determination of Ascorbic Acid

The ascorbic acid (vitamin C) of the chokeberry fruits was measured on day 0 and after 7 days of harvest, during their storage at 4 °C and 25 °C by macerating the sample mechanically with an establishing agent (5% metaphosphoric acid) and titrating the filtered extract against 2,6 dichlorophenolindophenol [37].

### 4.7. Determination of Anthocyanins, Total Phenolic Content, and Total Antioxidant Capacity

#### 4.7.1. Calculation of Concentration of Anthocyanin

An amount of 1 mL of the sample solution should be diluted with potassium chloride buffer and sodium acetate buffer, respectively, so that the absorbance readings at 530 nm are <1.2. After equilibrating the solution in the dark for 30 min, the absorbance of the sample at 520 and 700 nm was measured by spectrophotometer with a 1 cm cuvette calibrated with distilled water as the blank.

The difference in absorbance between pH values and wavelengths can be calculated as follows:Anthocyanin pigment (cyanidin-3-glucoside equivalents, mg/L) = A × MW × DF × 10 ^1^/ε × 1

The concentration of anthocyanin pigment can be obtained, where MW is the molecular weight; DF is the dilution factor; ε is the molar absorptivity derived from a full wavelength scan; 1 is for a standard 1 cm path length [38].

#### 4.7.2. Calculation of Concentration of Total Phenolics

Total phenolic content was determined by the Folin–Ciocalteu method, as was described by [39,40], with some modifications. The solution of the reaction consists of 2400 μL Folin–Ciocalteu (1:10 *v*/*v*), 80% (*v*/*v*) methanolic extract (100 μL), and nanopure water (500 μL). These chemicals were combined in tubes and then mixed via magnetic stirring. The mixture was allowed to react for 3 min, and then 2 mL of Na_2_CO_3_ (7.5% *w*/*v*) solution was added and mixed well. The solution was incubated at 37 °C for 5 min. The phenolic compounds in the sample are oxidized using the Folin–Ciocalteu reagent. The tubes were left to be cooled at room temperature (23 °C). The absorbance was measured at 760 nm using a spectrophotometer, and the results were expressed in gallic acid equivalents (GAE; mg/100 g dry mass) using a gallic acid standard curve.

#### 4.7.3. Calculation of Concentration of Antioxidant Capacity

For antioxidant capacity, fresh chokeberry fruits were accurately weighed (0.1 g) and cut into small pieces before extraction material was put in a mortar. After one second, they were mixed with 1 mL of 80% aqueous methanol. The homogenates were centrifuged at 12,000 rpm at 4 °C for 20 min [41]. For non-enzymatic antioxidant determination, the ability of the fruits to act as hydrogen or electron donors in the transformation of 1,1-diphenyl-2-picrylhydrazyl (DPPH) into its reduced form DPPH-H was measured. Total antioxidant activity was determined following the method of [42]. Fruit extract (50 μL) was added to 0.1 mM DPPH solution (2.95 mL). After 1 h, the absorbance of the reaction mixture was measured in triplicate at 517 nm on a spectrophotometer. The control solution was prepared by adding absolute ethanol (50 μL) to the DPPH solution. Measurements were expressed as scavenging activity %. The antioxidant activity [43] was determined by the following formula:Scavenging Activity (%) = {(Abs control − Abs sample)/Abs control} × 100
where Abs is the absorbance at 517 nm.

### 4.8. Mineral Analysis (ICP-OES)

Leaves and fruits of all treatments after the harvest on day 0 were gently cleaned with a mild detergent solution, shaken to remove excess water, and immediately rinsed thoroughly in tap water and distilled water. The samples were dried in an oven at 60 °C until constant weight and ground with a Wiley mill to pass through a 1 mm mesh screen before mineral concentration analysis. A total of 0.5 g sub-samples were then ashed at 500 °C for 4 h. The ash was dissolved in 6 N hydrochloric acid (HCl), filtered, and analyzed for macronutrients [phosphorus (P), potassium (K), calcium (Ca), and magnesium (Mg)] measured by Perkin Elmer Optima 8300DV ICP-OES.

### 4.9. Statistical Data Analysis

A complete randomized factorial design was used with four replicated per biostimulant treatment. Data were subjected to analysis of variance (ANOVA). For comparison of means, the Duncan multiple range test was used (*p* ≤ 0.05) using the SPSS 24.0 statistical package (SPSS, Inc., Chicago, IL, USA).

## 5. Conclusions

This study demonstrates the effectiveness of specific biostimulant treatments in improving the nutritional status and plant benefits of *Aronia melanocarpa* L. Quality traits and nutritional components depend on several factors, such as pre-harvest treatments and post-harvest storage conditions. The use of biostimulants, particularly the combination of the three (Mgl + Cpl + Kpl120), significantly improved the overall performance and, mainly, the uptake and accumulation of the macronutrients (P, K, Ca, Mg) in both leaves and berries and their quality characteristics of *Aronia melanocarpa* L. Biostimulants show a synergistic effect on nutrient availability and transport within the plant.

The increase in potassium and magnesium, essential for photosynthesis and stress resistance, suggests that biostimulant treatments enhance the plant’s physiological functions. The elevated calcium levels in the berries may also contribute to better fruit quality and longevity. Also, the elevated levels of key nutrients such as potassium and calcium, which are important for osmotic regulation and cell wall stability, suggest that these biostimulants may help *Aronia melanocarpa* L. plants better cope with abiotic stress, such as drought or temperature extremes. Future studies could explore the long-term impacts of these treatments under various environmental conditions to further validate their benefits.

## 6. Future Prospects

The biostimulant industry is rapidly growing, driven by increasing demand for sustainable agriculture. Ongoing research is expanding the understanding of how biostimulants work and identifying new sources and formulations that could further improve their efficacy and cost-effectiveness. The integration of biostimulants with precision agriculture technologies also holds promise for more targeted and efficient applications.

In summary, biostimulants represent a promising tool for enhancing plant growth, improving crop quality, and promoting sustainable agricultural practices. Their role in modern agriculture is likely to expand as farmers and researchers continue to explore and refine their use.

## Figures and Tables

**Figure 1 plants-13-03014-f001:**
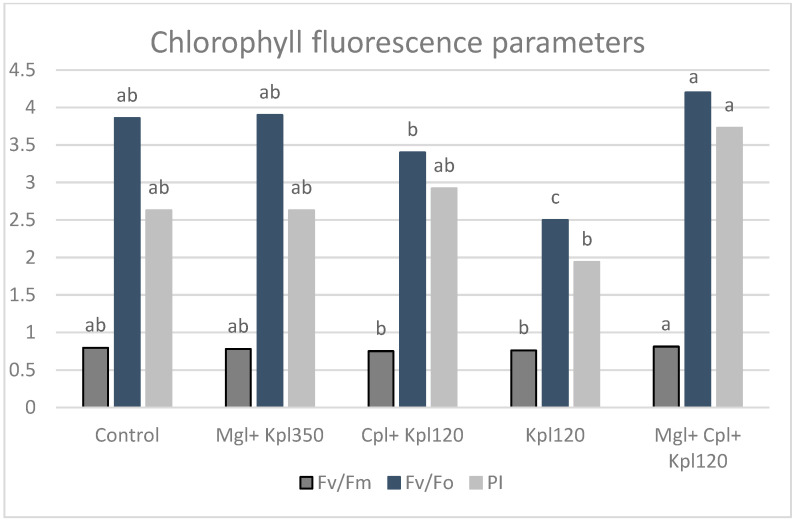
Chlorophyll fluorescence parameters (Fv/Fm, maximum efficiency of PSII photochemistry, Fv/Fo, efficiency of the oxygen-evolving complex on the donor side of PSII and PI, performance index) of black chokeberry leaves after biostimulants treatment (mean value, n = 4). Different letters for each parameter indicate significant differences according to Duncan’s multiple range test (*p* = 0.05).

**Figure 2 plants-13-03014-f002:**
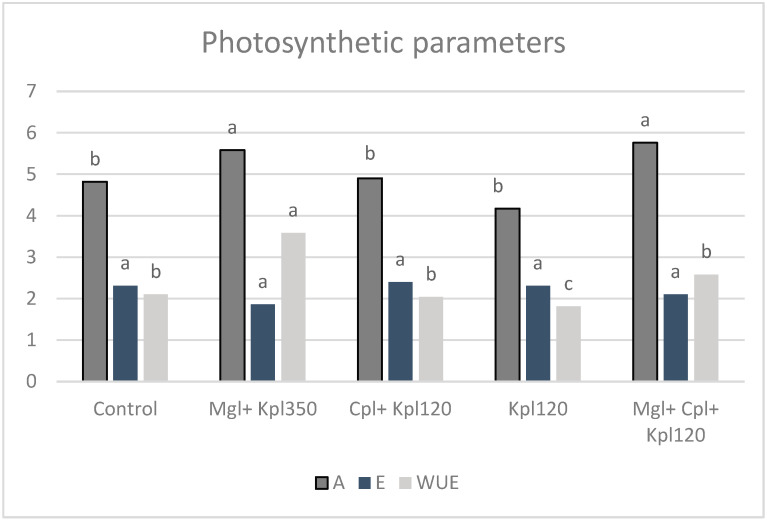
Photosynthetic parameters A (CO_2_ assimilation rate, (μmol CO_2_ m^−2^s^−1^)), E (Transpiration rate (mmol H_2_O m^−2^s^−1^)), and WUE (A/E, (μmol CO_2_/mmol H_2_O)) of black chokeberry leaves after biostimulants treatment (mean value n = 4). Different letters for each parameter indicate significant differences according to Duncan’s multiple range test (*p* = 0.05).

**Figure 3 plants-13-03014-f003:**
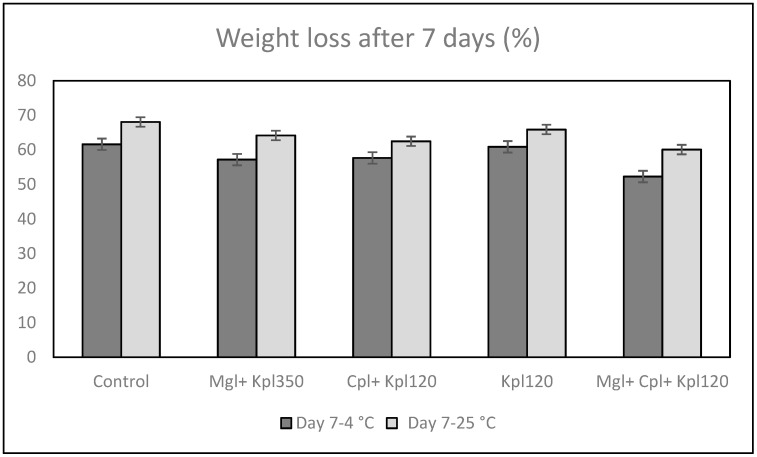
Weight loss rate (%) of black chokeberry fruits after biostimulant treatment, 7 days of harvest, and storage at 4 or 25 °C. Mean value, n = 10.

**Figure 4 plants-13-03014-f004:**
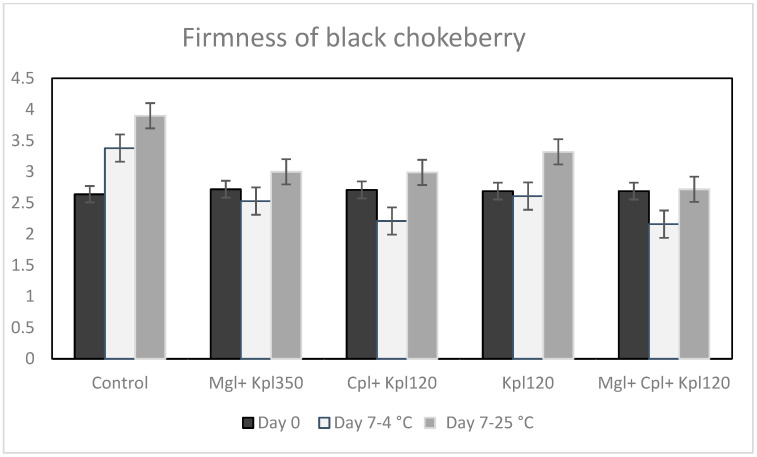
Flesh resistance in Newtons of black chokeberry fruits after biostimulant treatment, 7 days of harvest, and storage at 4 or 25 °C. Mean value, n = 9.

**Figure 5 plants-13-03014-f005:**
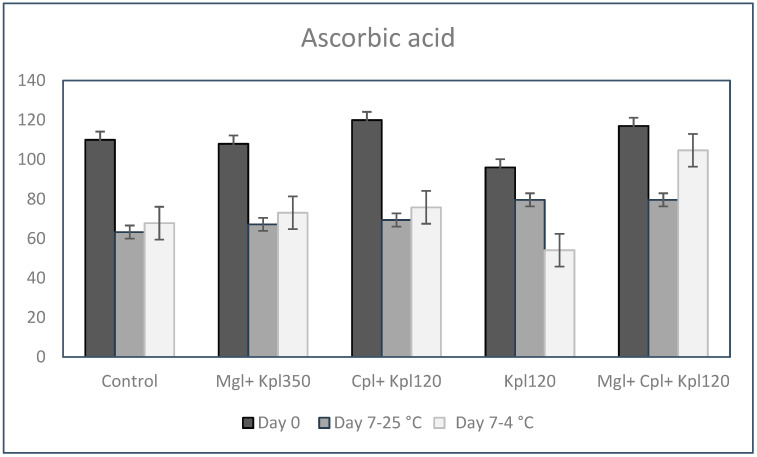
Ascorbic acid concentration in (units) AsA of black chokeberry fruits after biostimulants treatment, on day 0, and after 7 days of fruit harvest and storage at 4 or 25 °C. Mean value, n = 3.

**Figure 6 plants-13-03014-f006:**
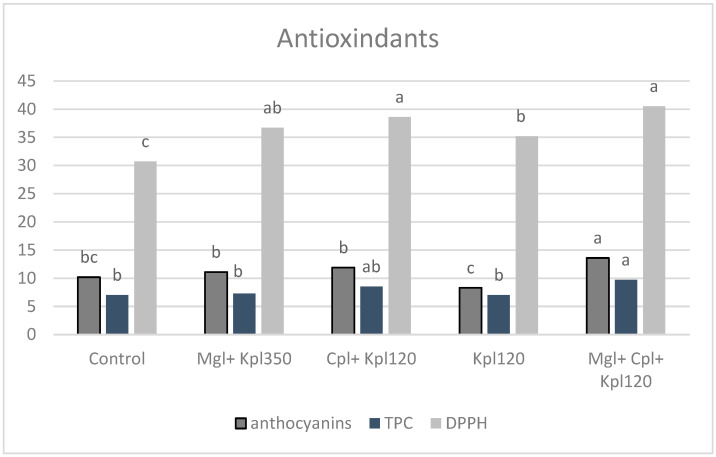
Antioxidant contents (Anthocyanins, total phenolic content, and 1,1-diphenyl-2-picrylhydrazyl (DPPH)-Total Antioxidant Capacity) of black chokeberry fruits after biostimulants treatment at day 0 (mean value n = 3). Different letters for each parameter indicate significant differences according to Duncan’s multiple range test (*p* = 0.05).

**Figure 7 plants-13-03014-f007:**
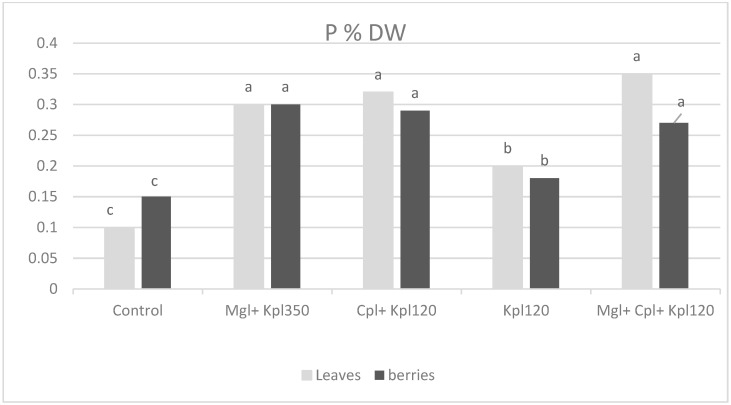
Phosphorus concentration (% dry weight) in leaves and berries for chokeberry plants after biostimulant treatment. Mean value, n = 3. Different letters for each parameter indicate significant differences according to Duncan’s multiple range test (*p* = 0.05).

**Figure 8 plants-13-03014-f008:**
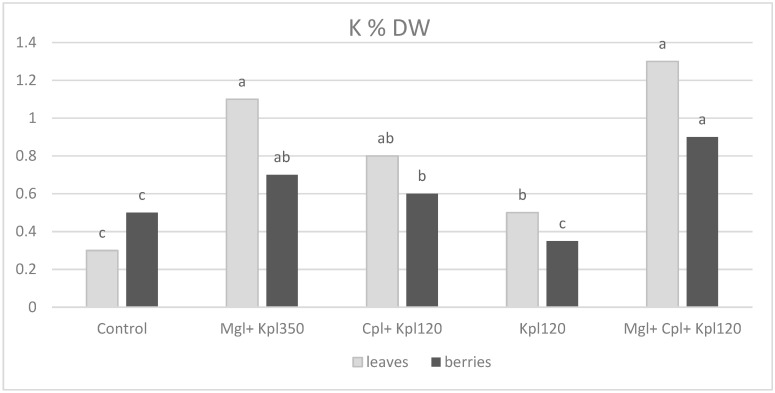
Potassium concentration (% dry weight) in leaves and berries for chokeberry plants after biostimulant treatment. Mean value, n = 3. Different letters for each parameter indicate significant differences according to Duncan’s multiple range test (*p* = 0.05).

**Figure 9 plants-13-03014-f009:**
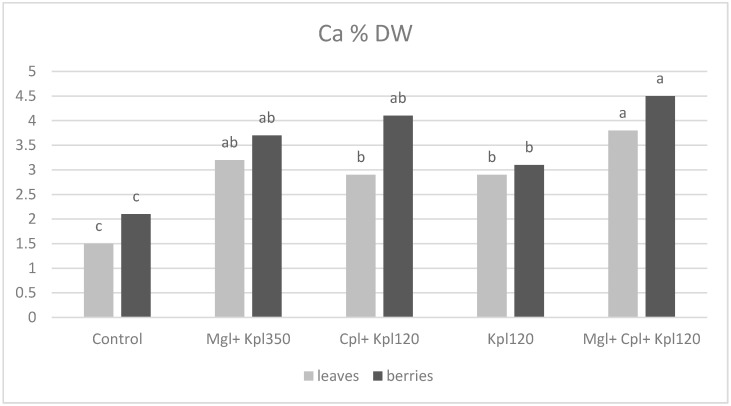
Calcium concentration (% dry weight) in leaves and berries for chokeberry plants after biostimulant treatment. Mean value, n = 3. Different letters for each parameter indicate significant differences according to Duncan’s multiple range test (*p* = 0.05).

**Figure 10 plants-13-03014-f010:**
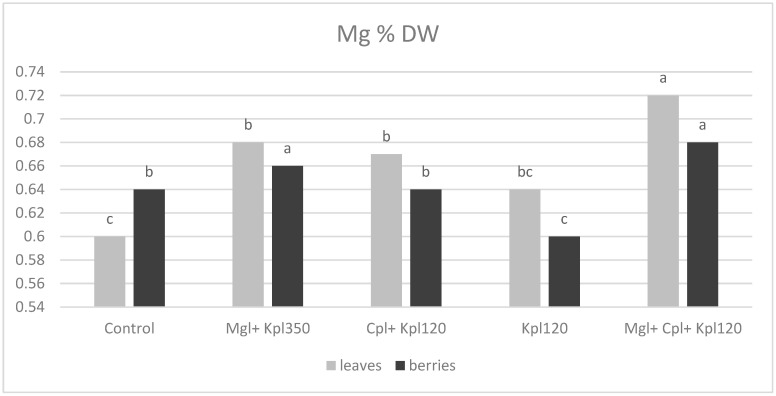
Magnesium concentration (% dry weight) in leaves and berries for chokeberry plants after biostimulant treatment. Mean value, n = 3. Different letters for each parameter indicate significant differences according to Duncan’s multiple range test (*p* = 0.05).

**Table 1 plants-13-03014-t001:** Treatments and biostimulants concentrations used for *Aronia melanocarpa* L. growth.

		Biostimulants		
Treatments	Magnablue^®^	Keyplex 120^®^	Keyplex 350^®^	Cropbiolife^®^
1st control (H_2_O)	X	X	X	X
2nd (Mgl + Kpl350)	2 mL/lt	X	2 mL/lt	X
3rd (Cpl + Kpl120)	X	0.5 mL/lt	X	2 mL/lt
4th (Kpl120)	X	0.5 mL/lt	X	X
5th (Mgl + Cpl + Kpl120)	2 mL/lt	0.5 mL/lt	X	2 mL/lt

## Data Availability

Data are available upon request.

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
