# Peer review of "Effects of Biostimulants on the Eco-Physiological Traits and Fruit Quality of Black Chokeberry (Aronia melanocarpa L.)"

_plants, 2024, doi:10.3390/plants13213014_

Round 1
Reviewer 1 Report
Comments and Suggestions for Authors
The paper covers a fascinating topic about plant biostimulants on an important shrub species.
Several ecophysiological measurements were measured during this experiment including gas exchange.
I have some minor comments and suggestions before accepting the paper for publication in Plants MDPI:
1) The abstract is too long and should be shortened. I would suggested the authors to report the differences between treatments in percentage to be clearer for the readers of Plants MDPI.
2) The hypothesis at the end of the introduction is missing and should be inserted. What was the current paper's novel aspect compared to the actual scientific literature?
3) I urge the authors to insert a full characterization of the used biostimulants this will help in the discussion part.
4) The discussion should be focused on the mode of action of the biostimulants and this could be facilitated based on the characterization of the biostimulants and what was the main signalling molecules which triggered the beneficial effects.
5) I urge the authors to run a Principal component analysis.
Author Response
Dear reviewer,
Thank you very much for your feedback on my paper
We have tried to make all the changes that were needed
Please find attached the corrections for our manuscript
Thank you for all the support

Reviewer 2 Report
Comments and Suggestions for Authors
Dear Authors,
thank you for the opportunity to meet the manuscript entitled: "The effects of biostimulants on the eco-physiological characteristics of black chokeberry plants (Aronia melanocarpa L.) and their fruit quality".
In this study, the authors investigated the effect of several biostimulati preparations on the physiological, production and quality parameters of aronia. The application of biostimulants is currently a very popular topic in the context of climate change, reduction in the use of chemical preparations and sustainable production. I therefore evaluate the choice of topic positively, but I have reservations about the preparation of the manuscript.
If pilot research was presented, a short-term study would also be acceptable. However, if research with commercial preparations is presented, it is desirable to present multi-year research. The weather conditions of the year always have the most significant influence on the parameters monitored. Therefore, it is difficult to consider one-year results as objective.
The trend is also to compare different commercial preparations. However, a detailed description of the preparations is also critical in this case. Detailed information on the composition of applied biostimulants is an absolute requirement for replicability and comparison of the results obtained by you with similar studies. Even if commercial products were used, in the scientific studies for the aforementioned, there must be detailed information about all ingredients or active substances, as well as the amount that was applied. I consider it absolutely insufficient if only general information about the type of substances is mentioned, without detailed characteristics.
There is no information about how the biostimulants were applied and when, or under what conditions.
Considering that commercial preparations were monitored in the manuscript, it would be appropriate to evaluate the economic efficiency. After all, it can be expected that the more nutrients and biostimulating substances you use, the more it will affect the results. However, the economics of such cultivation is questionable.
I recommend putting information about treatments (L291-296) into a table.
L28-29 "The pre-harvest manipulations with biostimulants..."; Incorrect wording. Based on this information, it could be assumed that biostimulants were applied just before the harvest and this had an impact on the mentioned things. In my opinion, it is enough to simply mention that "Biostimulants had a beneficial effect...".
You often use the conjunction "during the..." in the manuscript (e.g. L32; 250...). But "depending on" or something like that would be better.
In terms of keywords, I recommend avoiding those already mentioned in the title of the manuscript.
L202 How did you research this? Just because it's common knowledge doesn't mean you can put it in this manuscript without real research.
L241-242 Rhizobacteria were not included in any of the preparations, so this information is irrelevant to your research.
Author Response
Dear reviewer
Thank you very much for the corrections that you have proposed for our paper They were really very helpful
We have attached all the corrections point by point
Thank you very much for all the support

Reviewer 3 Report
Comments and Suggestions for Authors
The effects of biostimulants on the eco-physiological characteristics of black chokeberry plants (Aronia melanocarpa L.) and their fruit quality
The title should be more interesting for the readers and related to the most important problem in the work.
The abstract needs substantial revision. It is too long and consists of many meaningless words and phrases. The units should be revised. How do you use the biostimulants?
Introduction: Line 39 – correct, please. Check please the font, italics in the botanical names. What about this biostimulants, used in the research? Why do you measured such parameters and no others? What about it? The hypotheses of the study should be explicit intensively.
Results. You use the abbreviations without explanation. The same problem is in the tables and Figures, which should be read separately from the text. Units!
Fifure 2. Photosynthetic parameters? Which? Units!
Figure 3, 5: why do you use the line between combinations in the Figure? Don’t you consider the changes in the time period? Rebuild, please.
Texture – I don’t understand this part. How do you really measure firmness? Firmness is a meaningless word. Units, more information.
Figure 6. The capital letters?
Figure 7. What is ,,P concentration”? Fig. 8, 9, 10 the same. The figures should be made clear.
The discussion should be orderly. The results should not be repeated but interpreted.
Mateial and Methods. More clear. May be the table is indicated to the explanation of combination or figure. Producers of the preparation are needed.
4.4. 4.5. 4.6. Where are the results?
Formulas should be described according to the journal requirements.
Conclusion should be more scientific.
The biostimulant industry is rapidly growing from ca 50 years. This part is unnecessary. Some parts may be included in the Introduction or Discussion with the citation.
Comments on the Quality of English Language
Minor revision is required.
Author Response
Dear editor
Thank you for your valuable feedback
I appreciate your support throughout the revision process
We have made all the corrections and we have answer all the questions that the reviewers proposed As you can see in the attachment we have a point by point response for every reviewer's question.
Thank you for all the support
We hope now that our paper is appropriate for your journal

Round 2
Reviewer 3 Report
Comments and Suggestions for Authors
No comments. Thanks to the Authors for the corrections.